# Computed Tomographic Features of Thymus in Dogs: Correlation with Age, Gender, Breed and Body Fat Content

**DOI:** 10.3390/vetsci10070418

**Published:** 2023-06-28

**Authors:** Mohammad Molazem, Sarang Soroori, Alireza Bahonar, Saghar Karimi

**Affiliations:** 1Faculty of Veterinary Medicine, Department of Veterinary Surgery and Diagnostic Imaging, University of Tehran, Azadi Street, Tehran 1419963111, Iran; mmolazem@ut.ac.ir; 2Faculty of Veterinary Medicine, Department of Food Hygiene and Quality, University of Tehran, Azadi Street, Tehran 1419963111, Iran; abahonar@ut.ac.ir

**Keywords:** CT scan, thymic characteristics, canine, density, volume, grade

## Abstract

**Simple Summary:**

The thymus is the first lymphoid organ formed to regulate a newborn’s immunity. It reaches its maximum size during puberty, after which it undergoes an atrophic procedure called involution, but it was observed in humans that its ability to grow again in response to some stresses is maintained. There is no comprehensive study on computed tomographic features of thymus in dogs as a useful modality to evaluate the thoracic organs so this is our aim in the present study. According to our results, there is wide variation in thymic characteristics at different ages and there are some differences between breeds and genders. The present work can provide valuable information for veterinary literature.

**Abstract:**

Background: The thymus is the first lymphoid organ formed to regulate a newborn’s immunity. It reaches its maximum size during puberty, after which it undergoes an atrophic procedure called involution, but its ability to grow again in response to some stresses, such as infections, neoplasia, surgeries, chemotherapy, and radiotherapy is maintained. There is no comprehensive study on computed tomographic features of thymus in dogs. So, the goal of the present study is to gain better insight into the thymus using computed tomography as a non-invasive method. Methods: One hundred and fifty dogs classified in five age groups and five breed groups were recruited to this study and the thymus was evaluated using a 2-slice computed tomography machine. The inclusion criteria for the present study were having a normal complete blood count, plain and post-contrast CT scan examination of the thoracic region and no history of neoplasia, chemotherapy or radiotherapy. The visibility, density, enhancement, grade, size, volume, shape, borders and lateralization of the thymus were evaluated and statistical analysis was performed. The effect of obesity on thymic grade and volume was also investigated. Results: The visibility, density, dorsal length, volume and grade decreased with increasing age. The thymic shape and lateralization were mostly wedge shaped and left sided, respectively. The borders became concave with aging and increasing body fat content caused an increase in the fatty degeneration of the thymus. Conclusions: Declining thymic density, grade, size and volume with aging are related to thymic involution and fatty degeneration was accelerated by increasing body fat content. Females and males were different only in thymic shape and small and large breeds were different only in thymic volume. The thymus was visible in some geriatric dogs with no underlying disease. We expect that the present work can be used by radiologists in reading thoracic computed tomography but investigation of thymic characteristics in dogs with neoplasia and history of chemotherapy, radiotherapy and thoracic surgeries can complete this study.

## 1. Introduction

The thymus is the first lymphoid organ formed to regulate newborns’ immunity [1,2,3,4]. It reaches its maximum size during puberty, after which it undergoes an atrophic procedure called involution or degeneration, but its ability to grow again in response to some stresses, such as infections, neoplasia, surgeries, chemotherapy, and radiotherapy, is maintained [2,5]. The thymus is a bilobed lymphoid organ with a light pink color in puppies which turns into a light grey color in adult dogs [4,6]. It has an important role in growth and maturation of the immune system especially T cells [3,7]. Anatomically, it is laterally compressed and lies in the cranioventral part of the thoracic cavity in dogs [4]. This organ starts to involute with the changing teeth in dogs. Although the process is initially rapid, the tissue usually does not atrophy completely, even in old age. When it decreases in size and loses its lymphoid structure, it is replaced by fat. However, evidence of thymic tissue can be seen in most dogs, regardless of age [4]. Histologically, thymic shrinkage consists of the emptying of cortical lymphocytes and the substitution of adipose tissue and thymic corpuscles [8].

There are different conditions of the thymus, such as aplasia, hypoplasia, juvenile thymic hematoma, true thymic hyperplasia, rebound hyperplasia, thymic cysts, and thymic tumors such as thymoma and lymphoma [3]. Thymoma and thymic lymphoma are the most common thymic tumors explained in dogs [9,10,11]. Heart-based tumors, ectopic thyroid or parathyroid tumors, thymic branchial cysts, chemodectomas (aortic or carotid body tumors), metastatic neoplasia, granulomas, abscesses, lipomas, and teratomas must also be considered as differential diagnoses [12]. In humans, there is wide variation in thymic characteristics at different ages, making it important for radiologists to reduce unnecessary invasive procedures [2,5]. The recognition of a normal thymus on computed tomography (CT) is easier with knowledge of this normal variation, thereby avoiding false-positive diagnoses of mediastinal neoplasms [2,3]. So, the present study aims to describe the characteristics of the normal thymus in dogs using computed tomography as a non-invasive method and, to the best of our knowledge, this is the first study that evaluates the normal thymus specifically in dogs.

## 2. Materials and Methods

This descriptive and prospective study recruited a total of 150 dogs, 74 females and 76 males, from February 2022 until February 2023. The criteria for inclusion in this study were having a normal complete blood count (CBC), a plain and post-contrast CT scan examination of the thoracic region, and referring for CT scan examination with no thymic abnormality, no neoplasia in the thymus or other organs, and metastasis detection. These patients were referred for computed tomography due to traumatic injuries and orthopedic reasons. All medical records and CT scan images were also evaluated to exclude patients with any pathologies that have a possible effect on the thymus, such as lung nodules or lesions, sternal, mediastinal, or perihilar lymphadenopathy. The patients were of different breeds (Mix breeds, Pomeranian, German Shepherd, Shih Tzu, Maltese, Yorkshire, Terrier, Rottweiler, Spitz, Dachshund, Lhasa Apso, Jack Russel Terrier, Chihuahua, Grate Dane, Poodle, Border Collie, Pekingese, and Cocker Spaniel). They were divided into five age groups, with 30 dogs in each group, according to their ages. Group 1 was a puppy group, including puppies from ages 0 to 6 months old; group 2 was a juvenile group, including 7–12 months old; group 3 was young adults, including 1–2-year-old dogs; group 4 was an old adult group, including dogs aged 3–7 years old; and group 5 was a geriatric group, including dogs more than 7 years old. They were also divided into five size groups based on their size (twenty toy breeds, fifty small breeds, one medium-sized breed, sixty-six large breeds and thirteen giant breeds).

All CT examinations were performed by a 2-slice Somatom-Spirit CT scan machine (Siemens, Munich, Germany) at the department of Diagnostic Imaging of Veterinary Faculty, Tehran University, Tehran, Iran. The contrast media used for all examinations were Omnipaque (240 mg/mL) with a dosage of 4 mL/kg or Visipaque (320 mg/mL) with a dosage of 2 mL/kg. The standard protocol for thoracic CT scan was used (120 kVp, 60–65 mA, acquisition slice thickness of 1–1.5 mm, and reconstructed slice thickness of 1–3 mm). Dorsal and sagittal images were reconstructed and used for better detection of the thymus. All examinations were stored in the PACS system and then free version of radiant viewer was used to evaluating the images. The Leonardo Syngo workstation (Siemens, Germany) was also used for volumetry. Evaluations were performed by a board-certified radiologist in the Osteo window.

The thymus was evaluated in the first step to determine whether it was visible or not. Non-visible thymus was that with complete degeneration seen with fat attenuation and not distinguishable from mediastinal fat. After that, a grading system for thymic tissue appearance was used based on its predominant component (grade 0: totally fatty replacement of the thymus (non-visible) and could not differentiate from mediastinal fat; grade 1: more than 50% fatty replacement of the thymus; grade 2: more than 50% soft tissue component in the thymus; grade 3: totally soft tissue component in the thymus. After grading the thymus, the attenuation (CT number) of thymic tissue in plain and contrast-enhanced images was measured. For this purpose, a round-shaped region of interest (ROI) with a surface equal to 75 mm^2^ was considered in the widest part of the organ for large thymus (Figure 1) and a smaller ROI for small thymus. Then pre- and post-contrast Hounsfield Units (HU) were compared for each patient individually and also between groups for evaluating the pattern of attenuation alteration and enhancement alteration.

The size of the thymus was also measured in transverse and dorsal images. The maximum height (TH) and diameter (TD) in transverse images and maximum diameter (DD) and length (DL) in the dorsal views were measured (Figure 2), as well as thymic volume (TV) in all groups. These indices were measured in the area related to the mediastinal fat at the normal location of the thymus for patients with a completely fatty replacement of the thymus (grade 0). All measurements were performed in contrast-enhanced examinations. To normalize the thickness of the thymus in the transverse view, the height of the second thoracic vertebra midbody was measured and the ratio was calculated. The correlation between thymic size and grade with the body fat content was evaluated by measuring the thickness of subcutaneous fat in the dorsal aspect of the T4 spinous process (Figure 3). Subjective characteristics such as predominant deviation of the thymus to left, midline or right position, the thymic shape (wedge shaped, rectangular shaped, or linear shaped), and the thymic contour (convex and concave) were also evaluated. For patients with fatty degeneration, these characteristics were analyzed in mediastinal fat at the location of the previous thymus. Evaluation of thymic size, volume, attenuation, and grade was performed twice by a single person within a week, and the mean value of the first and second measurements for numerical data and the second results for thymic grading were reported.

### Statistical Analysis

Categorical data were collected as numbers and presented as a percent value. Continuous data were presented as the mean ± SD. Mean density, TD, TH, DD, DL, and TV were compared between groups using a one-way ANOVA and the Tukey method; the *t*-test was used for comparing numerical data between the two groups; and the Pearson Chi-square test was used to analyze the categorical variables such as enhancement, shape, predominant side, and grading. The Pearson correlation coefficient (r value) was calculated for the evaluation of the repeatability and reliability of two-stage measurements (intra-observer agreement). All statistical analyses were performed with SPSS software version 16.

## 3. Results

### 3.1. Visibility

In the present study, the thymus was visible in 104 out of 150 cases. The distinct lobes of the thymus were not distinguishable. The visible thymus was noted with a soft tissue component and homogenous appearance or a combination of soft tissue and fat component based on the degree of involution with heterogenous (mottled) appearance (Figure 4, Figure 5 and Figure 6). We accounted the thymic tissues with complete fat degeneration for non-visible thymus (Figure 7). The anatomical and topographical locations of the thymus are shown in Figure 4, Figure 5, Figure 6 and Figure 7. The thymus’ dorsal border begins at the ventral aspect of the vessels in the cranial mediastinum and ends at the dorsal aspect of the sternebra. In the dorsal and sagittal views, it was extended caudally up to the pericardial region and merged with the pericardium on the left side (Figure 2B). The visibility percentage in group 1 was 100%. The minimum and maximum ages in group 1 were three and six months, respectively. In groups 2 and 3, the thymus was visible in 86.7% of cases and it was not notable in 13.3% of cases. The minimum and maximum ages with visible thymus in group 2 were seven and 12 months, respectively. This group includes one 11-month-old and three 12-month-old cases with a non-visible thymus. In group 3, the youngest and oldest cases with a visible thymus were 1.5 and 2 years old. The patients with a non-visible thymus were two years old. In group 4 the visibility percentage was 53.3% and the thymus was not visible in 46.7% of patients. The youngest and oldest patients with visible thymus in group 4 were three- and six-year-old dogs. The visibility percentage of group 5 was 20% and the maximum age with visible thymus was nine years old (Table 1).

### 3.2. Density (Attenuation)

It was found that the mean density of the thymus decreased with increasing age from group 1 to group 5, but the pattern was not the same between the patients in one group with different ages and the distribution of CT numbers in one group was very wide. The mean, minimum, and maximum densities in plain and post-contrast CT scan examinations in each group were summarized in Table 2 and Table 3. The differences in mean density between plain and post-contrast CT scan examination regarding age groups were significant (*p*-value ≤ 0.05, Table 2 and Table 3). The mean density difference between each group (for example, group 1 with other groups) was evaluated and summarized in Table 4. There was a significant difference between the mean density of group 1 with other groups. Further, there was a significant difference between the mean density of group 2 with groups 1, 4, and 5. The difference in the mean density of group 3 was significant compared to groups 1, 4, and 5, while the difference in the mean density of groups 4 and 5 was not significant. The Pearson correlation coefficient for intra-observer agreement of two-stage measurement of CT numbers was 84.5% and 85% for plain and post-contrast CT scan examinations, respectively (Table 2 and Table 3).

### 3.3. Enhancement

Ninety-four-point seven percent of the thymic tissues in this study were enhanced. According to the results, 0.01 percent of the visible thymus was not enhanced (two patients out of 104). The other non-enhanced thymic tissues were non-visible (six patients out of 46). The percentage of enhancement in groups 1, 2, and 4 was 100%, and in groups 3 and 5, it was 93.3% and 80%, respectively. The *p*-value for comparing enhancement showed a significant difference between age groups (Table 1, *p*-value ≤ 0.05). There was no significant difference between the thymic enhancement of different breeds and genders. The thymus was enhanced in 93% of females and 96% of males. It was also noted that the blood supply of the thymus was mostly by the left and right internal thoracic arteries and veins (Figure 4, Figure 5, Figure 6 and Figure 7).

### 3.4. Shape

Three different shapes of the thymus in the transverse section were noted in the present study: a rectangular shape with convex borders that was visible in younger patients, a wedge-shaped appearance with concave or straight borders in young or adult patients (Figure 4, Figure 5, Figure 6 and Figure 7), and a linear shape that was seen predominantly in older patients (Figure 8). The thymic shape in the sagittal view was variable. Categorization of thymic shape could not be established in the sagittal nor in the dorsal sections. In the dorsal view, the thymus was wider in the cranial and caudal portions and in the mid-part it was narrower and became narrower with aging in all patients (Figure 2B). The shape of the transverse section of the thymus was wedge shaped with a concave border at 74.4%, rectangular with a convex border at 22%, and linear at 3.3%. The difference in shape between age groups and different breeds was not significant, but there was a significant difference in the shape of the thymus between genders (*p*-value = 0.006). The percent values of each kind of shape are summarized in Table 1 and Table 5. The thymus shape in the only medium-sized breed in this study, a seven-year-old female Cocker Spaniel, was wedge shaped.

### 3.5. The Predominant Side

The dominant side of the thymus in the cranial portion was evaluated because the thymus deviated to the left side caudally in all patients, resulting in the well-known ‘sail sign’ in radiographic images. The predominant side of the thymus was left sided in 46% of the cases and mid-line in 53% of the cases. However, there was only one patient with a thymus deviated to the right side in the cranial aspect of the mediastinum (Figure 8). The predominant side percentage of each group was summarized in Table 1. The difference in the predominant side was significant between groups 1, 2, and 3. In group 1, the thymus was mostly in the midline and it was deviated to the left side in groups 2 and 3. There was no significant difference between breeds for this variable.

### 3.6. Volume

The mean, maximum, and minimum thymus volumes of each group are summarized in Table 6. The values of thymic volume decreased with aging, and there was a significant difference between age groups for this variable (*p*-value = 0.000). The mean volume of the thymus in females was 2.3 ± 2.6 cm^3^ and 2.5 ± 3.8 cm^3^ in males, with no significant difference between males and females, but a significant difference in volume between breeds. The mean values of volume in toy breeds, small breeds, large breeds and giant breeds were 1.4 cm^3^, 1.7 cm^3^, 2.9 cm^3^, and 3.6 cm^3^, respectively. The thymic volume in the only medium-sized breed patient was 2.3 cm^3^. The Pearson correlation coefficient was high between the 2-stage measurement of thymic volume (97.5%). There was no correlation between thymic volume and subcutaneous fat content (r = 0.053).

### 3.7. Diameter, Length, and Height

The mean, minimum, and maximum values of the thymic diameter and height in the transverse view and the diameter and length in the dorsal view are shown in Table 6. There is no significant difference in TD, TH, and DD between age groups, but the difference in DL was significant (*p*-value ≤ 0.05). The approximate ratio between TD and the height of the T2 midbody was 1.2 ± 0.7.

### 3.8. Grade

The grade of the thymus has been declining with increasing age. There was no grade 0 thymus in group 1. There was no grade 3 and 2 thymus in group 5. Based on our findings, the correlation between age and grade was significant. The percent value of each grade in each group is summarized in Table 1. Based on the correlation coefficient, there was a reverse correlation between grade and subcutaneous fat content (r: −0.18).

## 4. Discussion

The thymus was visible in all puppies, most of the juveniles and young adults, half of the old adults and some geriatric dogs. According to the results, the thymic component started to atrophy and be replaced by fat from the first months of life, and the fat content was increasing with aging. In one ultrasonographic study of the thoracic region in dogs and cats carried out by Reiche and Wisner (2000) [13], they concluded that the thymus degenerates after puberty. According to our results and the results of this ultrasonographic study, prepuberty period marks the onset of thymic involution but it speeds up after puberty. In one study of the humans, the thymus in 783 patients with no underlying disease was evaluated, and it was recognized that the thymus was present in 100% of patients under the age of 30, 73% of patients between the ages of 30 and 49, and 17% of patients over 49 years of age [14]. Hence, regarding thymus visibility, our results show the same trend as in humans, as it declines by aging.

We observed that the thymic soft tissue component can be preserved in some healthy adult and geriatric dogs. Evans and de Lahunta [4] mentioned that thymus does not atrophy completely and evidence of its remnant can be visible at any age. In humans, it was shown that some disorders, such as infection, tumorigenic conditions, and chronic inflammations accelerate the fatty involution of the thymus [7,15]. It was also shown that the thymus can regrow after recovery from the stress [2]. This phenomenon, which is called ‘rebound hyperplasia’, can happen some months after resolving the stress [2] but this is not confirmed yet in dogs and it is seen rarely in veterinary [16]. Cordella et al. (2023) [16] concluded that the thymus can be visible in adult dogs with different non-thymic neoplasia compared to young dogs and correlated this finding to thymic hyperplasia rather than neoplasia. So, if the thymus is visible with residual soft tissue component in adult and geriatric dogs and there are no clinical signs or suspicion of any neoplasia or infections, it will be a normal variant or it can be hyperplasia related to previous resolved stress. In a human study, Simanovsky et al. (2012) [5] found that soft tissue components in the thymus of four healthy patients aged 40–54, suggesting that the finding of some soft tissue in an asymptomatic patient was occasional, but could be normal. According to another human study, it seems that the shape of the visible thymus can help to differentiate hyperplasia from thymic abnormality. Based on this study by Sklair-Levy et al. (2000) [17], the biconvex margins late in the second decade of life or the focal expansion of a lobe or multilobulation at any age are usually indicative of thymic abnormality. Being aware of all of these, it seems that there is still lack of information about thymus changes in dogs at the time of abnormal conditions such as non-thymic neoplasia, surgeries, radiotherapy and chemotherapy; so, more investigations will be needed in the future.

We hypothesized that the early fatty degeneration of the thymus in some juveniles and young adults in our study can relate to some conditions, such as previous treated infections or inflammations which accelerated the thymic atrophy.

As noticed in the present study, the distinct lobes of the thymus were not seen distinctly in both cranial and caudal portion and our hypothesis was the low accuracy of the 2-slice CT scan machine; but in a most recent study by Cordella et al. (2023) [16] that evaluated the thymus in adult dogs with non-thymic neoplasia in comparison to thymus in young dogs (under 9 months) using a multidetector computed tomography, although they used a 320-row CT scan machine, the bilobed thymus was not noted except in one patient. Evans and de Lahunta [4] also mentioned previously, thymic lobes are not differentiable in cranial portion because of the connective tissue attaching both lobes in this part.

In our study, the same thymic topographical characteristics as Evans and de Lahunta’s study were observed [4]. From an anatomical point of view in a small dog, the caudal aspect of the left thymic lobe is placed between the thoracic wall and the left ventricle and the right lobe of the thymus that usually touches the cranial surface of the pericardial sac is extended laterally [4].

According to the present results, thymic appearance was homogenous with a predominant component of soft tissue, and became heterogenous with mottled appearance by adipose substitution. Cordella et al. (2023) [16] also observed that the thymus appearance was lobulated in adults and homogenous in young dogs. In fact, the combination of soft tissue and fat component makes this lobulated or mottled appearance in thymus.

The CT attenuations of thymus in the present study were in the range of soft tissue and fat based on the predominant component, but there was a wide spectrum of densities in each group and between individuals. CT numbers noted in our study for each patient matched the subjective grade. Declining CT numbers from puppies to geriatric dogs is also due to fat replacement by aging. In fact, there is a reverse correlation between thymus CT attenuation and aging which was also noticed in dogs according to Cordella et al. (2023) [16] and human studies [5].

According to the present findings, only in eight patients, which were young adults and geriatric, no enhancement was notable. In six of these patients, the thymus was also non-visible (grade 0); so non-enhancement could be due to severe atrophy of the thymus. Another reason can be the very small size of the degenerated thymus, which reduces the accuracy of density measurement. Another hypothesis is insufficient injected contrast media, but when the enhancement of other organs was checked, this hypothesis was ruled out. The blood supply of the thymus was by left and right internal thoracic arteries based on our results, although according to Evans and de Lahunta [4], occasionally a thymic branch from brachiocephalic trunk on the right side and subclavian artery on the left side may supply this organ. Evaluating the thymus vasculature in other populations may add more information about thymic blood supply.

The shape of the thymus in the transverse section was variable between dogs, even in the same age and breed group, but we classified the shape. According to our results, it was observed that most of the thymic shapes in the transverse section were wedge shaped with straight and concave borders, perhaps because most of the patients were adults. Opposite of our expectations, most of the puppies’ thymus were wedge shaped with convex border and there were some with rectangular shape and convex border. We hypothesized that this finding was due to premature thymic involution following previous stresses or diseases in puppies. Alteration of thymic borders from convex to straight and concave with aging was also confirmed in human studies [18].

The thymic shape was also significantly different between genders. Females had more rectangular-shaped thymus than males. Contrary to our presumption old adult dogs had more rectangular-shaped thymus than juveniles and young adults; it can be related to the body conformation of dogs in this group. Another reason can be related to the fact that there were more male dogs in the juvenile and young adult groups. The thymic shape was so variable in the sagittal view, making it impossible to classify it in this view. Thymic shape in the dorsal view was the same in all patients. According to Cordella et al. (2023) [16] who evaluate the thymic shape in the dorsal view, the thymus was triangular and elongated in most young and adult patients although there was no shape classification in their study.

The dominant side of the thymus was also variable, but in all of the samples, the caudal aspect of the thymus extends to the left side, which makes a ‘sail sign’ in ventrodorsal thoracic radiographs. This feature in humans is right-sided and the sail sign is notable on the right side in anterior-posterior radiographs [2]. The dominant side was significantly different between puppies, juveniles, and young adults. The thymus was mostly in the midline in puppies and deviated to the left side with age, but this pattern was not repeated in old adults and geriatrics, thus this variable needs more investigation in the future. The predominant side of the thymus was studied by Cordella et al. (2023) [16] and they observed that thymus lateralization was mostly left sided in adults and midline in young dogs, and the same as our finding, they noticed only one right-sided thymus. In one human study evaluating the side predominancy of the thymus, it was also centered at the midline or showed a left-side predominance in most patients [5].

In the present study, the transverse view was mostly used to evaluate the characteristics because it is the main non-reconstructed view with better resolution and because of the best categorization of features such as thymic shape and side predominancy in this view.

In our study population, the volume and thymic length in the dorsal view decreased with aging. The mean values of each index of the thymus were measured with a high reliability coefficient, but there was a wide variation in thymic size between groups and there was no descending pattern in some measurements from puppies to geriatrics. We hypothesized that this finding is due to breed and weight differences. Cordella et al. (2023) [16] also observed a variation in thymic dimensions between different patients with different body weights.

Anatomically, in beagles, it is known that when the thymus is completely developed it is 12 cm long, 6 cm wide, and 3 cm high [4]. Moarrabi et al. (2019) [19] performed radiographic and ultrasonographic examinations of the thymus in five Mongrel puppies every month from birth to five months of age. They reported the thymic width and length at five months of age were about 6.56 ± 0.63 mm and 28.10 ± 2.72 mm, respectively, which were close to the reported measurements of our study.

It was observed that the fat replacement of the thymus increases with an increment in subcutaneous fat content, which can be an index of obesity and body condition in our population. The impact of obesity on the thymic attenuation in human patients between 20 and 30 years old was evaluated in one study that showed overweight patients were more likely to have higher levels of thymic fatty replacement than normal or underweight patients [14]. In the same way Araki et al. (2016) [20] showed that patients with low thymic grades had significantly higher body condition scores.

There are many comprehensive studies on human’s thymus characteristics using imaging modalities which provide valuable data about this organ in newborns, children and adults. Based on these investigations, thymus CT attenuation in plain and contrast-enhanced examinations is variable [2,18]. In all of these studies, the thymic CT attenuation declines with aging due to thymic involution and fat replacement [2,3,5,14].

Ackman et al. (2016) [20] described differences in thymic density between the genders in the 20–30 age groups and they showed increased fatty degeneration of the thymus in young males compared to females [21]. In our study, there is no difference between all males and females in thymic density and the amount of fat replacement. Simanovsky et al. (2012) [5] found no correlation between gender and thymic volume and density in humans.

As we noticed, there was a great variation in thymic shape in dogs, there was also significant variation in the CT scan appearance of a normal thymus in the preadolescent and adolescent age groups according to Heiberg et al. (1982) [18] that investigated the normal and abnormal thymus using CT scan examinations in 40 human cases under 20 years old. They also found that the thymus is larger in younger humans, but the thymic size varied widely in all ages. Thymic size and dimensions were also so diverse between individuals in the present study.

Variation of normal thymus appearance on human CT scans and differences in thymic shape between males and females in the middle and older age was also reported by Araki et al. (2016) [20] which evaluated the impact of physical characteristics such as body mass index (BMI) and cigarette smoking on thymus shape in middle and old-age human patients. Their findings suggested that the start of thymic atrophy in females is delayed by approximately 10 to 20 years in comparison to males. Thymic atrophy is completed before the age of 60 in men and 70 in women, and sexual steroids are considered to affect the thymic shape [20]. We do not report similar findings in our study.

A study by Hussain et al. (2020) [22] investigated how diverse imaging modalities can be used to assess thymus normality and abnormality to differentiate thymic hyperplasia from thymic malignancy in human patients. According to their study, the CT scan is valuable for primary assessment, providing the morphology and density, and MRI is the imaging modality of choice in diseased thymus because of its ability to provide a functional assessment of this organ.

The important limitation in the present study was the 2-slice CT scanner, which causes loss of some details and a decrease in resolution. Human studies for thymus CT scans were performed by different kinds of CT scan machines with different slices [15]. Multidetector row CT (MDCT) scanners can image the thymus with more detail because of thinner slices, unceasing images, and multiplanar reconstructions [5]. Another limitation of the present study was the lack of histopathologic procedures following CT scan examinations to confirm the thymic tissue.

## 5. Conclusions

To the best of our knowledge, this is the first study evaluating the normal thymus in dogs using computed tomography and performing classification and grading. As a conclusion, variable shapes, sizes, and densities of the thymus can be noted in dogs, although some patterns for alteration of these variables with aging are explained through this study. The ratio of thymus thickness in the transverse view to the height of the second thoracic vertebral midbody was approximately 1.2 ± 0.7. A grading system for thymic appearance based on predominant thymic components was also established through this study and can be used by radiologists for reading thoracic CT scans. The thymus was also visible in adult and geriatric dogs with no underlying non-thymic disease, but due to lack of sufficient data in this field, serial examinations were still suggested in the case of suspicions of infiltrative diseases in old dogs owing some remnant thymic tissue. According to the results, thymic atrophy did not significantly affect enhancement. Thymic characteristics were not significantly different between males and females except in the shape of the thymus, which needs more investigation to generalize this finding to other populations. Small and large breed dogs were different only in the volume of the thymus. Further, it can be concluded that increasing body fat content affects thymic fatty degeneration, which can influence the immunity. Finally, further investigation of thymic alterations in the case of infiltrative disease and some stresses such as chemotherapy and radiotherapy can complete the present study.

## Figures and Tables

**Figure 1 vetsci-10-00418-f001:**
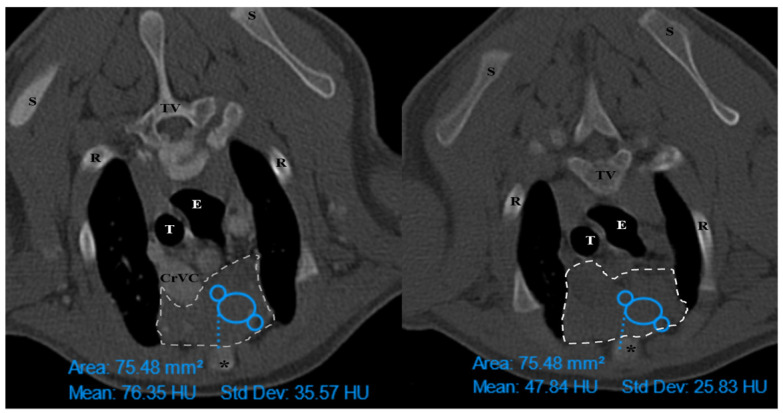
Illustration of density measurement of thymus in plain (**right**) and post-contrast (**left**) CT scan examination in a ROI equal to 75 mm^2^. These figures belong to a four-month mixed-breed male puppy. The grey dash lines show the grade3 rectangular-shaped midline-position thymus. S: scapula, TV: thoracic vertebra, R: rib, E: esophagus (which is dilated due to general anesthesia), T: trachea, CrVC: cranial vena cava, and asterisk: second sternebra.

**Figure 2 vetsci-10-00418-f002:**
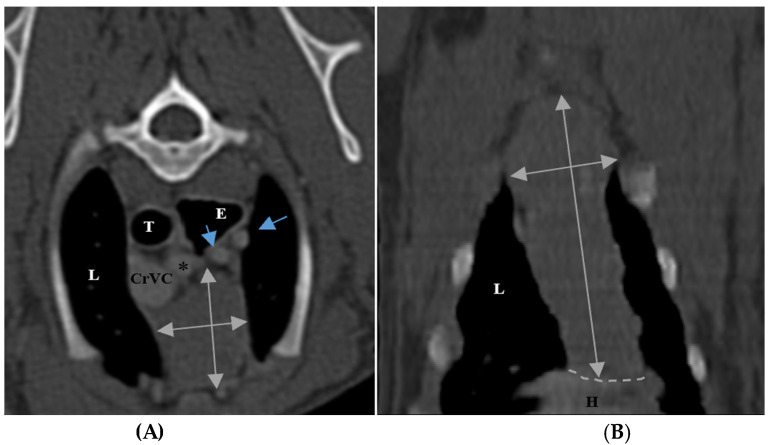
Measurement of maximum height and diameter of thymus in transverse section (**A**) and maximum length and diameter in dorsal section (**B**) in post-contrast CT scans. The grey double arrows show the regions of measurement. Both figures belong to a five-month-old female shih Tzu. The thymus is grade 3, rectangular shaped with one convex border and midline position. L: lung, T: trachea, E: esophagus (which is dilated due to general anesthesia), CrVC: cranial vena cava, H: heart, black asterisk: right subclavian artery, shorter blue arrow: left common carotid artery, and longer blue arrow: left subclavian artery.

**Figure 3 vetsci-10-00418-f003:**
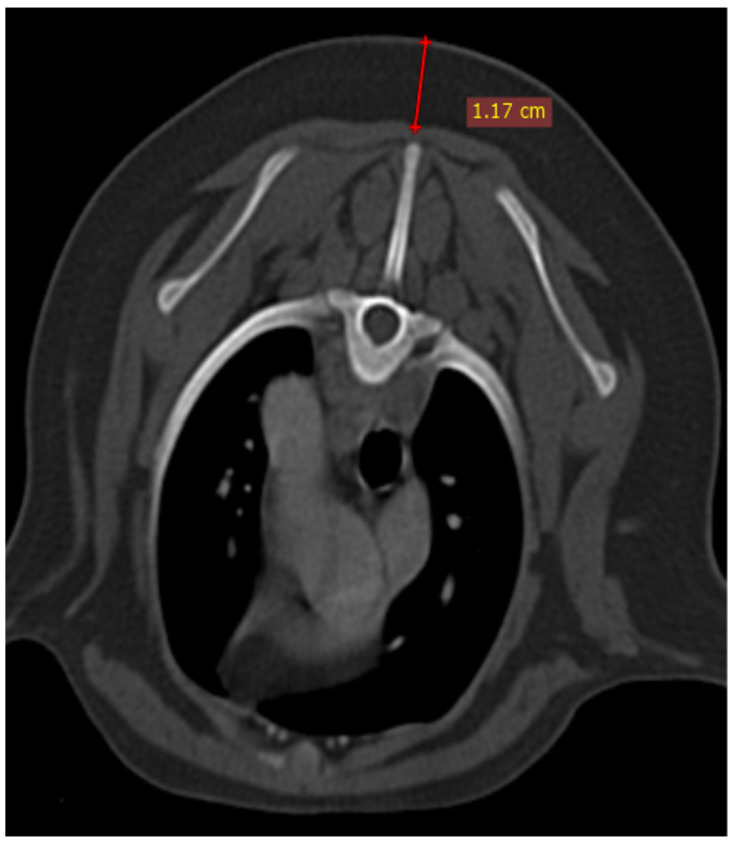
Demonstration of measurement of subcutaneous fat content at the level of T4 in a transverse section of post-contrast CT examination.

**Figure 4 vetsci-10-00418-f004:**
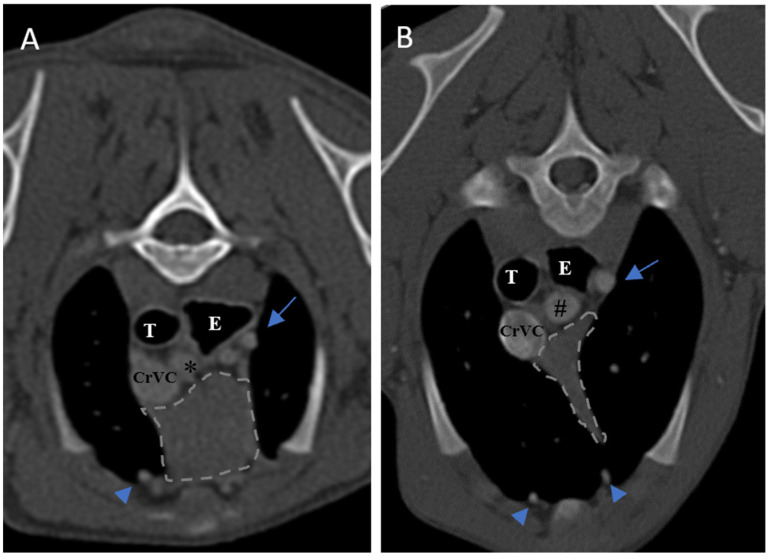
Demonstration of visibility and topographic location of grade 3 thymus. (**A**,**B**) are contrast-enhanced transverse sections of the thoracic region at the level of T2 and T3, respectively. (**A**): Five-month-old female Shih Tzu with a visible, homogenous, grade 3, rectangular-shaped and midline predominancy thymus with CT number equal to 93. (**B**): Four-month-old male mixed-breed dog with a visible, homogenous, grade 3, wedge-shaped and left-sided predominancy thymus with CT number equal to 90. Light grey dash lines illustrate the thymic borders. T: trachea, E: esophagus (which is dilated due to general anesthesia), CrVC: cranial vena cava, asterisk: right common carotid artery, hash sign: brachiocephalic trunk, arrows: left subclavian artery, and arrowheads: internal thoracic arteries.

**Figure 5 vetsci-10-00418-f005:**
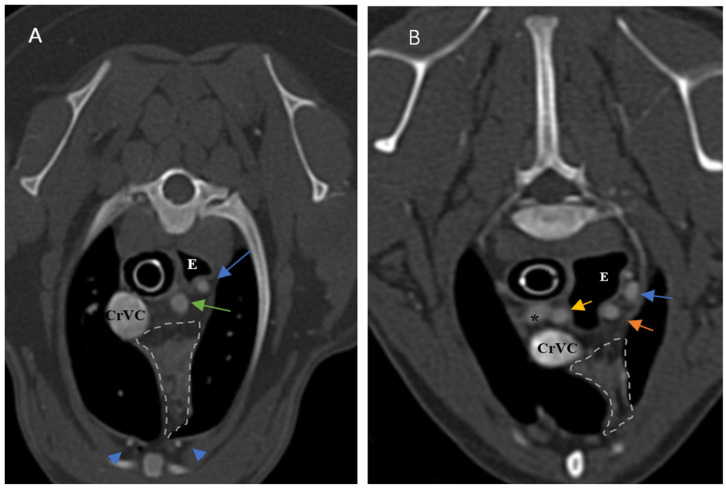
Demonstration of visibility and topographic location of grade 2 thymus in contrast-enhanced transverse sections of the thoracic region at the level of T3 in both patients. (**A**): One-year-old female mixed-breed dog with a visible, grade 2, wedge-shaped thymus with midline predominancy, mottled appearance and concave border (CT number: 75). (**B**): Two-year-old female mixed-breed dog with a visible, grade 2, wedge-shaped thymus with left-sided predominancy, mottled appearance and concave border (CT number: 79). Light grey dash lines illustrate thymic borders. CrVC: cranial vena cava, E: esophagus (which is dilated due to general anesthesia), blue arrow: left subclavian artery, green arrow: brachiocephalic trunk, blue arrowheads: internal thoracic arteries, yellow arrow: right common carotid artery, orange arrow: left common carotid artery, and asterisk: right subclavian artery.

**Figure 6 vetsci-10-00418-f006:**
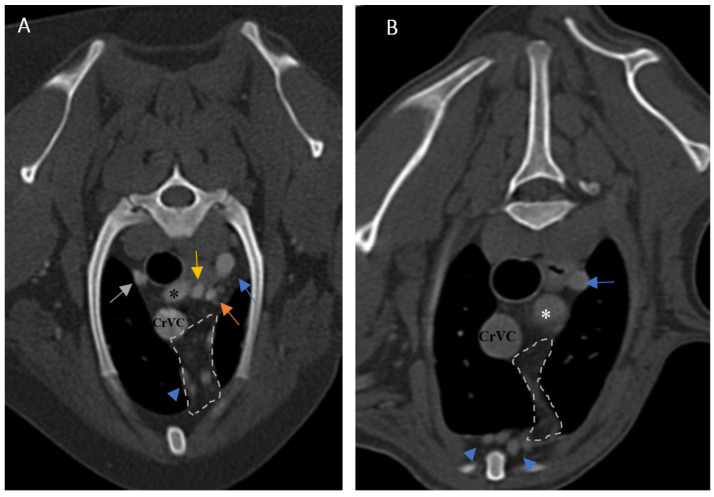
Illustration of visibility and topographic location of grade 1 thymus in contrast-enhanced transverse sections of the thoracic region at the level of T3 (**A**) and T4 (**B**). (**A**): Six-year-old female Terrier with a visible, grade 1, rectangular-shaped thymus with concave borders, left-sided predominancy and mottled appearance (CT number: −55). (**B**): Nine-year-old male terrier with a visible grade 1, rectangular-shaped thymus with concave borders, left sided predominancy and mottled appearance. Light grey dash lines show thymic borders. CrVC: cranial vena cava, grey arrow: right costocervical vein, blue arrow: left subclavian artery, yellow arrow: right common carotid artery, orange arrow: left common carotid artery, arrowheads: internal thoracic arteries and veins, black asterisk: right common carotid artery, and white asterisk: brachiocephalic trunk.

**Figure 7 vetsci-10-00418-f007:**
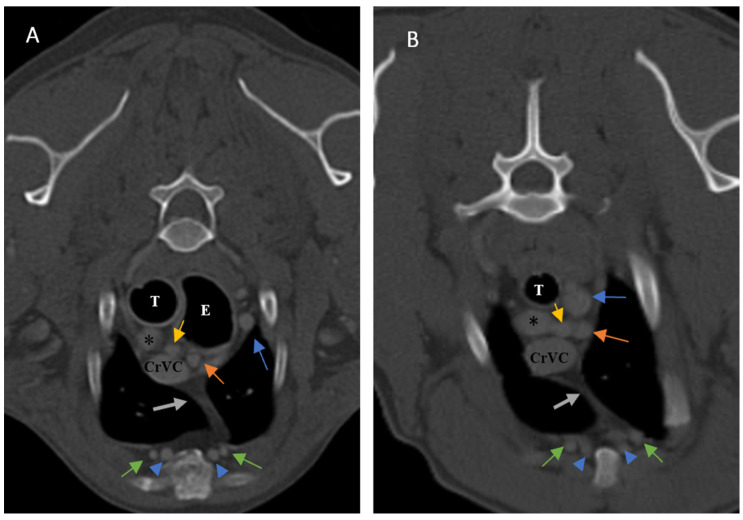
Illustration of non-visible grade 0 thymus and topographic location of thymus in contrast-enhanced transverse sections of the thoracic region CT scan examination at the level of T2. (**A**): Ten-year-old female mixed-breed dog. The thymus is grade 0 (non-visible), wedge shaped with homogenous appearance, left-sided predominancy and straight borders (CT number: −137). (**B**): One-year-old female mixed-breed dog. The thymus is grade 0 (non-visible) and wedge shaped with homogenous appearance, left sided-predominancy and concave borders (CT number: −156). CrVC: cranial vena cava, T: trachea, E: esophagus (which is dilated due to general anesthesia), grey arrows: degenerated thymus, blue arrows: left subclavian artery, yellow arrows: right common carotid, orange arrows: left common carotid artery, green arrows: internal thoracic artery, arrowhead: internal thoracic veins, and asterisks: right subclavian artery.

**Figure 8 vetsci-10-00418-f008:**
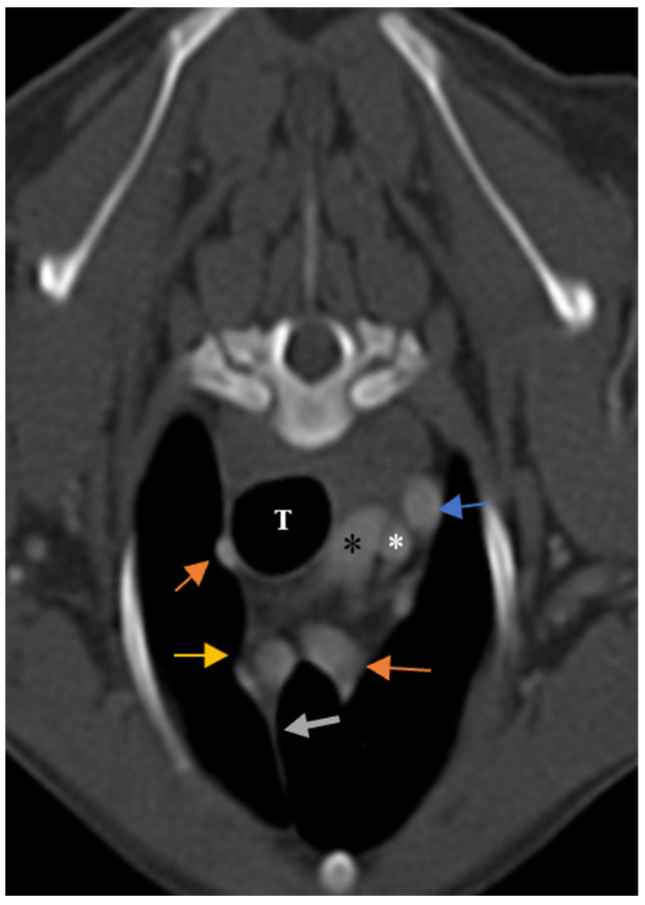
The only right-sided thymus (grey arrow) in the present study which is also completely involuted in a six-year-old male poodle. The degenerated thymus which is not distinguishable from mediastinal fat has a linear shape. T: trachea, blue arrow: left subclavian artery, short orange arrow: left costocervical vein, long orange arrow: left subclavian vein, yellow arrow: right subclavian vein, black asterisk: brachiocephalic trunk, and white asterisk: left common carotid artery.

**Table 1 vetsci-10-00418-t001:** Percent values of each categorical variable of the thymus at different ages and *p* values.

Age Group (*n* = 30)	Group 1	Group 2	Group 3	Group 4	Group 5	*p* Value ≤ 0.05
Visibility (%)	100	86.7	86.7	53.3	20	
Enhanced (%)	100	100	93.3	100	80	0.001
Shape						≤0.05
Wedge shape (%)	63.3	76.7	86.7	66.7	80	
Rectangular (%)	36.7	23.3	6.7	33.3	10	
Linear (%)	0	0	6.7	0	10	
Side predominancy						0.016
Left-sided (%)	43.3	50	70	40	26.7	
Mid-line (%)	56.7	50	30	60	73.3	
Grade (%)						0.000
3	66.7	20	6.7	0	0	
2	23.3	53.3	46.7	6.7	0	
1	10	20	36.7	53.3	20	
0	0	6.7	10	40	80	

**Table 2 vetsci-10-00418-t002:** Mean, Minimum, and maximum densities in plain CT scan examination. The *p* value and r value are also demonstrated.

Age Groups	Mean	Minimum	Maximum	*p* Value (≤0.05)	r Value (%)
Group 1	24.3 ± 39	−40	60		
Group 2	−2.2 ± 34	−145	63		
Group 3	−11.6 ± 39	−160	80		
Group 4	−84.1 ± 36	−180	−5		
Group 5	−100 ± 22	−140	−41		
Total				0.000	84.5

**Table 3 vetsci-10-00418-t003:** Mean, minimum and maximum densities in contrast-enhanced CT scan examination. The *p* value and r value are also demonstrated.

Age Groups	Mean	Minimum	Maximum	*p* Value (≤0.05)	r Value (%)
Group 1	62.1 ± 29	−14	135		
Group 2	35 ± 41	−80	144		
Group 3	20.06 ± 42	−109	100		
Group 4	−45.2 ± 37	−140	2		
Group 5	−74.3 ± 34	−139	−16		
Total				0.000	85

**Table 4 vetsci-10-00418-t004:** *p* values from comparing mean densities between groups (*p* value ≤ 0.05).

	Group 1	Group 2	Group 3	Group 4	Group 5
Group 1	-	0.019	0.000	0.000	0.000
Group 2	0.019	-	0.8	0.000	0.000
Group 3	0.000	0.8	-	0.000	0.000
Group 4	0.000	0.000	0.000	-	0.3
Group 5	0.000	0.000	0.000	0.3	-

**Table 5 vetsci-10-00418-t005:** Percent values of different shapes of thymus based on gender and breed.

Shape of Thymus	Wedge	Rectangular	Linear
Female	66.2%	32.4%	1.4%
Male	82.9%	11.8%	5.3%
Toy Breeds	60%	35%	5%
Small Breeds	70%	24%	6%
Large Breeds	80.3%	18.2%	1.5%
Giant Breeds	84.6%	15.4%	0%

**Table 6 vetsci-10-00418-t006:** The mean, maximum, and minimum values of numerical measurement of the thymus in age groups.

Groups	Group 1	Group 2	Group 3	Group 4	Group 5	*p* Value	r Value (%)
TD (mm)						0.1	79
Mean	10.5 ± 6.8	9.4 ± 4.8	7.3 ± 4.8	8.3 ± 4.9	8.9 ± 3.1		
Max	34	20.5	13	30	16.5		
Min	4.1	2	3	2.8	4.5		
TH (mm)						0.083	75
Mean	21.3 ± 9.2	20.7 ± 7.9	18.4 ± 7.3	17.1 ± 7.4	16.2 ± 9.8		
Max	48.5	43	41.5	33	43		
Min	4.2	11.5	10	2.5	5		
DD (mm)						0.061	79
Mean	9.1 ± 6.5	7.7 ± 3.4	5.8 ± 2.8	7.4 ± 4.7	6.7 ± 3.07		
Max	32	14.5	14.5	27	14		
Min	4.5	2.8	1	12	1		
DL (mm)						0.02	72
Mean	27.6 ± 12.1	28.6 ± 13.7	25.8 ± 6.9	23.8 ± 8.05	20.8 ± 8.4		
Max	50	58	40	42	42.5		
Min	9.05	7.3	15	8.5	9		
TV (cm^3^)							
Mean	4.7 ± 5.5	3.03 ± 3.3	1.7 ± 1.2	1.3 ± 1.1	1.2 ± 0.9		
Max	24.8	16.8	6.05	4.5	4.2		
Min	0.19	0.18	0.12	0.03	0.27		

## Data Availability

Not applicable.

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
