# Peer review of "Computed Tomographic Features of Thymus in Dogs: Correlation with Age, Gender, Breed and Body Fat Content"

_vetsci, 2023, doi:10.3390/vetsci10070418_

Round 1
Reviewer 1 Report
The authors bring interesting information about the tomographic findings of the thymus in dogs.
However, the methodology needs to be improved mainly in the division of groups. The authors divide the size of the animals into two groups - large and small, but dogs such as the Cocker spaniel are considered medium-sized or medium-sized. Up to a weight of 15 kilograms is considered for medium and not large breeds. See in the literature the weight range for medium, large and giant breeds
you will have to review all data and results and discussion after adapting the methodology groups, referring to race. and table 5.
To improve image description - add arrows and other markers to make images self-explanatory
line 350 - 352 it's not discussion - improve
Furthermore, the article is well written.
Sincerely
Minor editing of English language required
Author Response
Dear Reviewer
Thank you for nice commenting and points. We improve the introduction and add more references to improve the study. In the case of breed classification, we revised our classification based on literature and then revised the results and discussion although the results didn't have significant changes after revision of breed classification but your comment about this point was completely right and we appreciate that. The discussion section was also improved to be more comparative. The images were revised and some new images with more markers and labels were added to the study. Again thank you for your comments. We hope our revision will be satisfactory. We will be also ready if additional revisions will be needed. There is also a cover letter which will be attached here.
Thank you in advance
Saghar Karimi

Reviewer 2 Report
the study is conceived correctly and adequately written; it may have a clinical usefulness and be a starting point for future developments. I have no further specific comments to offer.
Author Response
Dear reviewer
Thank for your kind comments. We appreciate your time to review our manuscript. We hope that we can provide useful information for veterinary literature. There is also a cover letter in attachment part.
Thank you so much
Saghar Karimi

Reviewer 3 Report
The present work is an interesting analysis of the changes in the thymus morphology of dogs, according to age and sex, using computed tomography.
However, your manuscript must be improved. All the comments appear in the attached file.

The quality of the English is acceptable, but there are some details pointed out in the attached file that need to be corrected
Author Response
Dear Reviewer
So many thanks for nice commenting. We really appreciate your time for review our article. We revised all the writing faults in the text based on the comments. The introduction section improved by adding more details about thymus anatomy and function as you commented. The results were also revised. The tables were placed in their correct place and the values were revised based on comments. About the the values of group 3 in table 5, we added a comment in the revised file. More pictures with more markers and labels were added to result section. The discussion section were also developed by changing the format to be more comparative as your comment. The references were revised based on your comments and 3 more references were added. We hope that the revision form will be satisfactory. We will be ready for additional revision if needed. The cover letter is also attached here.
Thank you in advance
Saghar Karimi

Round 2
Reviewer 1 Report
Thanks for the invitation to review this manuscript. The changes made were important for the quality of the manuscript.
Author Response
Dear reviewer
Thanks again for your time and nice commenting. We hope that the present manuscript will provide useful data for veterinary literature.
Saghar Karimi
Reviewer 3 Report
Dear Authors,
Your manuscript has been significantly improved. However, there are still some little mistakes that should be corrected.
All the comments are in the attached file.

Regarding English editing, your manuscript needs to be revised by a native speaker
Author Response
Dear reviewer
Thank you again for your time and nice commenting. We tried to apply all your comments and points as mentioned bellow and we hope that it will be satisfactory:
- We revised all the writing errors in the text based on your comments.
- The references in the text also were revised based on the comments.
- Table 1 was also revised .
- We changed the key words.
- About figure 7B, we added this case because we want to show a grade-0 thymus in young adult dog which can be an exception.
We hope that the revision will be satisfactory but we will be again ready for any needed revision.
Thank you in advance
Saghar Karimi